# Antibiotic Resistance, Virulence Gene Detection, and Biofilm Formation in *Aeromonas* spp. Isolated from Fish and Humans in Egypt

**DOI:** 10.3390/biology12030421

**Published:** 2023-03-10

**Authors:** Dalia El-Hossary, Asmaa Mahdy, Eman Y. T. Elariny, Ahmed Askora, Abdallah M. A. Merwad, Taisir Saber, Hesham Dahshan, Nora Y. Hakami, Rehab A. Ibrahim

**Affiliations:** 1Medical Microbiology and Immunology Department, Faculty of Medicine, Zagazig University, Zagazig 44519, Egypt; 2Department of Botany and Microbiology, Faculty of Science, Zagazig University, Zagazig 44519, Egypt; 3Department of Zoonoses, Faculty of Veterinary Medicine, Zagazig University, Zagazig 44519, Egypt; 4Department of Clinical Laboratory Sciences, College of Applied Medical Sciences, Taif University, P.O. Box 11099, Taif 21944, Saudi Arabia; 5Department of Veterinary Public Health, Faculty of Veterinary Medicine, Zagazig University, Zagazig 44519, Egypt; 6Department of Medical Laboratory Technology, Faculty of Applied Medical Sciences, King Abdulaziz University, Jeddah 21589, Saudi Arabia

**Keywords:** prevalence, *Aeromonas* spp., haemolysin, aerolysin, antimicrobial resistance, multiple antibiotic resistance index, PCR, microtiter plate assay

## Abstract

**Simple Summary:**

Despite significant advancements in food safety and cleanliness, there is still a risk of food supply chain contamination. Fish are susceptible to infection from bacterial pathogens, especially *Aeromonas* spp. *Aeromonas* spp. are dangerous to people because they can spread illnesses like septic arthritis, gastroenteritis with diarrhea, skin and soft tissue infections, meningitis, and bacteremia through the consumption of infected fish. Therefore, the present study was conducted to detect the prevalence of, antibiotic resistance, virulence, and biofilm formation by *Aeromonas* spp. in raw fish markets and humans in Zagazig, Egypt. In the current study, 11 isolates were confirmed as *Aeromonas* spp. and 4 isolates were confirmed as *Aeromonas hydrophila* using biochemical PCR assays. In addition, the antimicrobial resistance profile of the *Aeromonas* isolates was tested against 16 antibiotics, the result of which indicates the susceptibility of all isolates to imipenem followed by chloramphenicol, and a high multiple antibiotic resistance (MAR) index range between 0.142–0.642 was detected. In addition, it was illustrated that two isolates of the four identified *A. hydrophila* isolates were positive for the *aer* gene and that one isolate only had the *hly* gene. Additionally, biofilm formation was detected in two isolates from tilapia muscles and mugil viscera. Based on the information provided, control measures might be implemented to stop the high-risk contamination of a specific area and protect individuals from multidrug-resistant strains that might be transmitted through the food chain or incorrect handling.

**Abstract:**

The genus *Aeromonas* is widely distributed in aquatic environments and is recognized as a potential human pathogen. Some *Aeromonas* species are able to cause a wide spectrum of diseases, mainly gastroenteritis, skin and soft-tissue infections, bacteremia, and sepsis. The aim of the current study was to determine the prevalence of *Aeromonas* spp. in raw fish markets and humans in Zagazig, Egypt; identify the factors that contribute to virulence; determine the isolates’ profile of antibiotic resistance; and to elucidate the ability of *Aeromonas* spp. to form biofilms. The examined samples included fish tissues and organs from tilapia (*Oreochromis niloticus*, n = 160) and mugil (*Mugil cephalus*, n = 105), and human skin swabs (n = 51) and fecal samples (n = 27). Based on biochemical and PCR assays, 11 isolates (3.2%) were confirmed as *Aeromonas* spp. and four isolates (1.2%) were confirmed as *A. hydrophila*. The virulence genes including haemolysin (*hyl A*) and aerolysin (*aer*) were detected using PCR in *A. hydrophila* in percentages of 25% and 50%, respectively. The antimicrobial resistance of *Aeromonas* spp. was assessed against 14 antibiotics comprising six classes. The resistance to cefixime (81.8%) and tobramycin (45.4%) was observed. The multiple antibiotic resistance (MAR) index ranged between 0.142–0.642 with 64.2% of the isolates having MAR values equal to 0.642. Biofilm formation capacity was assessed using a microtiter plate assay, and two isolates (18.1%) were classified as biofilm producers. This study establishes a baseline for monitoring and controlling the multidrug-resistant *Aeromonas* spp. and especially *A. hydrophila* in marine foods consumed in our country to protect humans and animals.

## 1. Introduction

Egyptian aquaculture is crucial to the country’s food supply. In particular, Nile tilapia and mugil aquaculture are crucial to Egypt’s food supply [1]. Fish has a significant role in human nutrition, contributing at least 20% of the protein consumed by one third of the world’s population [2]. Despite these fascinating features, fish are prone to bacterial pathogen infection, which causes significant economic losses in many nations [3,4]. These pathogenic bacteria have been linked to serious fish disorders such as motile aeromonad septicemia, bacterial gill infection, and tail and fin rot [5]. One of the most prevalent bacteria that infects fish with infectious illnesses is *Aeromonas hydrophila* [6,7].

*Aeromonas* spp. are significant contributors to normal *microbiota* in freshwater and saltwater environments. They also pose a threat to humans because they can cause dangerous diseases like septic arthritis, gastroenteritis with diarrhea, skin and soft tissue infections, meningitis, and bacteremia [8,9]. Aeromonas hydrophila transmits the zoonotic disease aeromoniasis to humans through the intake of tainted fish and water [10]. Other potential disease-causing pathogens include Aeromonas caviae and Aeromonas sobria, alongside A. hydrophila, which is thought to be the primary culprit infecting fish and aquatic organisms [11]. These Aeromonas spp. produce motile aeromonad septicemia (MAS) in fish, which manifests symptoms such as exophthalmia, erosion, scale removal, and ulceration [6]. There are several virulence factors that are connected to *Aeromonas*’*s* pathogenicity, including haemolysins, which in turn are divided immunologically into extracellular haemolysins and aerolysins, cytotoxic enterotoxins, extracellular polysaccharides, proteases, lipases, and biofilm formation [12]. These elements greatly influence the emergence of illnesses in both fish and humans. Additionally, the primary virulence factors linked to gastroenteritis infection in humans also include cytotoxic heat-labile enterotoxin (Act), cytotonic heat-stable enterotoxin (Ast), and cytotonic heat-labile enterotoxin (Alt).

In addition to serving as growth promoters, antibiotics are crucial in the treatment of illnesses in aquatic environments. However, the misuse of these antibiotics has resulted in long-lasting leftovers in aquatic habitats and soil sediments. In turn, this leads to antibiotic resistance [13]. The capability of bacteria to withstand a variety of these antimicrobials enhances additional virulence factors, resulting in the appearance of multiple forms of antibiotic resistance (MAR) in *Aeromonas* spp. [14,15], which may be transmitted to humans and animals through the consumption of contaminated fish products [16,17].

Biofilm formation is the process of bacterial cells adhering to biotic or abiotic surfaces and encasing themselves in a polyanionic, hydrated matrix made up of exopolymeric materials (EPSs), polysaccharides, proteins, and nucleic acids [18]. Owing to their ability to cling to a variety of surfaces and produce biofilms, *Aeromonas* species are thought to pose a risk to the public’s health, particularly to people who live near the coast [19]. Food deterioration, product rejection, financial losses, and all food-borne diseases are brought on by the development of food biofilms, which are the main source of contamination [20].

This study was carried out to provide up-to-date information about *Aeromonas* spp. in raw fish in Zagazig, Egypt, aiming to (i) isolate and identify *Aeromonas* spp. from 99 raw fish markets; (ii) identify the factors that contribute to their virulence; (iii) determine the isolates’ profile of antibiotic resistance; and (iv) elucidate the ability of *Aeromonas* spp. to form biofilms.

## 2. Material and Methods

### 2.1. Sample Collection

A total of 343 samples were collected for our study; out of the 343, 265 fish samples comprising 160 tilapia (*Oreochromis niloticus*) were collected as follows: 57 samples from viscera, 57 liver samples, and 46 muscle samples. A total of 105 mugil (*Mugil cephalus)* samples were collected, including 38 viscera samples, 25 liver samples, and 42 muscle samples from local fish markets in Egypt’s Sharkia Governorate. The fish samples were collected fresh with a mean average weight of 19 ± 1 g and a length of 10 ± 0.3 cm in sterilized polyethylene bags on ice and analyzed immediately after transporting them under aseptic conditions to the Microbiology lab. The fish samples were firstly enriched with APW (Merck, Germany) under shaking conditions at 37 °C for 24 h. According to the standards outlined by the international commissions on microbiological criteria for foods [21], the viscera and muscles of surface-sterilized fish samples were separated under totally aseptic circumstances. In addition, human samples including 51 hand swabs and 27 stool samples were obtained in sterile alkaline peptone water after being collected under aseptic circumstances (APW, Oxoid CM1028). To determine the prevalence of this pathogen in regularly ingested tilapia and mugil fish, from the university hospitals in Zagazig City, Egypt, stool samples from gastroenteric patients with watery diarrhea who occasionally or regularly consumed retail fish were collected. Hands of fish handlers were swabbed using sterilized swabs and then the swabs were collected in sterile alkaline peptone water tubes and immediately transported to the laboratory under aseptic conditions. Written informed consent has been obtained from the patient(s) to publish this paper.

### 2.2. Isolation and Identification of Aeromonas spp.

Twenty-five g of the collected samples, including fish tissues and organs from both tilapia and mugil and human skin swabs and fecal samples were enriched with alkaline peptone water (Merck, Germany) and incubated under shaking conditions at 37 °C for 24 h. After overnight incubation, serial dilutions were made, then 100 µL aliquots from these dilutions were streaked onto selective *Aeromonas* agar media (Himedia, Egypt), and the plates were incubated at 37 °C for 24 h under aerobic conditions. The colonies were then examined under a light microscope for a Gram stain and morphological characterization [22].

The obtained colonies were tested for catalase and oxidase activity, then characterized biochemically in the usual manner by loading 100 µL of the microbial suspension aseptically onto the analytical API 20 E kits (bioMerieux, France) according to the manufacturer’s instructions.

### 2.3. Molecular Identification and Virulence Genes Detection

The existence of *Aeromonas* genus- and species-specific genes was checked using a PCR for molecular identification. An extraction of bacterial DNA was performed by thermolysis according to the method described by DeSilva et al. [23]. An amount of 100 µL of overnight cultured broth was mixed with 400 µL of sterile distilled water in eppendorf tubes that were transferred to the heat block for 5 min at 95 °C, followed by centrifugation at 15,000 rpm for 2 min at 4 °C; the supernatant was used as DNA template and stored at −20 °C for PCR study. Negative controls (DNA-free) and positive control strains (provided by the AHRI, Dokki, Egypt) were included in the PCR assay. Reactions were performed in a total volume of 25 μL. Each volume containing, 5 μL of a 5X master mix (taql/high yield, Jena Bioscience, Jena, Germany), consisted of DNA polymerase, dNTPs mixture, (NH4) SO4, MgCl2, Tween 20, Nonidet P-40, stabilizers, 1.25 μL of each primer (forward and transverse) (20 pmol/μL), 5 μL of genomic DNA, and 12.5 μL of double deionized distilled water. The confirmed *Aeromonas* isolates were subjected to a PCR for the identification of A. hydrophila. The primers of the 16S rRNA gene used for amplification are illustrated in Table 1. The following PCR cycles were run in a thermal cycler: one cycle at 94 °C for 4 min, 30 cycles at 94 °C for 30 s, 60 °C for 30 s, and 72 °C for 30 s, and then a final extension at 72 °C for 7 min (Eppendorf, Germany). The PCR products were subjected to direct sequencing in both directions following purification using a PureLink PCR product purification kit (Thermo-Fisher Scientific, Bremen, Germany). Multiple alignments were performed on the obtained sequences. To enable visualization, the amplified PCR products were electrophoretically separated by 1.5% agarose gel (Applichem, Germany, GmbH). A gene ruler and a 1 Kb plus DNA ladder were used to measure the sizes of the amplified product (Fermentas, Thermo Scientific, Germany).

Additionally, the *hlyA* gene sequences were conventionally amplified by PCR to identify the hemolysis-associated gene determinants in bacterial isolates with the specific primers 5-CTA TGA AAA AAC TAA AAA TAA CTG-3 and 5-CAG TAT AAG TGG GGA AAT GGA AAG-3. The steps for the *hly A* and *aer A* genes’ amplification were performed in accordance with that the procedure described by Yousr et al. [26]. Primer pairs 5-CAC AGC CAA TAT GTC GGT GAA G-3 and 5-GTC ACC TTC TCG CTC AGG C-3 were used to analyze the *aer* gene.

### 2.4. Antimicrobial Susceptibility Test

Using the Kirby-Bauer disc diffusion method, the isolated *Aeromonas* spp. were evaluated for antibiotic susceptibility according to the procedure described by the Clinical and Laboratory Standards Institute [24], and the antibiotics were selected following the Clinical Laboratory Standards Institute’s recommendations (CLSI). The Muller–Hinton broth (Oxoid, CM0405) was inoculated with the isolated and purified *Aeromonas* inocula after the turbidity had been adjusted to a McFarland’s turbidity (2 × 10^8^ CFU/mL) of 0.5. Afterward, 100 µL of the turbid broth was plated over the Mueller–Hinton agar (MHA) surface where the antibiotic discs (Oxoid Limited, Basingstoke, Hampshire, UK) were placed and incubated at 37 °C for 24 h. ceftriaxone (CRO 30), cefotaxime (CTX 30), ceftazidime (CAZ 30), imipenem (IPM 10), gentamicin (CN 10), tetracycline (TE 30), and chloramphenicol (C) were interpreted according to Clinical and Laboratory Standards Institute (CLSI) guidelines [27], ciprofloxacin (CIP 5) was interpreted according to Clinical and Laboratory Standards Institute (CLSI) guidelines [28], and streptomycin (S 10) was interpreted according to CLSI guidelines [29]. In the case of cefixime (CFM 5), tobramycin (TOB 10), norfloxacin (NOR), and nalidixic acid (NA 30), the interpretation relied on the French Society of Microbiology [30]. Moreover, kanamycin (K 30) assessment was done according to the guidelines of Comite de l’Antibiogramme de la Societe Francaise de Microbiologie [31]. In addition, the multiple antibiotic resistance (MAR) index was calculated by dividing the number of antibiotics to which the bacteria were resistant by the total number of studied antibiotics [32].

### 2.5. Biofilm Formation Assay

The capability of isolated and purified *Aeromonas* bacteria to produce biofilms was examined using a modified version of the microtiter plate method [33]. An overnight culture of *Aeromonas* isolates was injected into 10 mL of Trypticase Soy Broth (TSB) (Merck, Germany) and incubated at 37 °C for 24 h. An amount of 200 µL of the 1:100 diluted overnight cultures were turbidity-adjusted to 0.5 McFarland in fresh TSB. Then, 100 µL of each diluted broth was inoculated into each of the microtiter plate wells and incubated at 37 °C for 24 h. The controls were made by inoculating the wells with only TSB without bacteria. The well’s contents were thrown away after being thoroughly cleaned with PBS in triplicate. After that, 200 µL of 1% (*w*/*v*) crystal violet dye was used to stain the plate, and it was left at room temperature for one hour. An amount of 200 µL of 95% (*v*/*v*) ethyl alcohol was added to the dried wells after being washed three times with PBS and incubated at 25 °C for 15 min. Thereafter, an ELISA (enzyme-linked immunosorbent assay) plate reader (Biotek SX2, Winooski, VT, USA) was used to measure the optical density at 600 nm. The experiment was performed in triplicate. The biofilm formation capability of *Aeromonas* isolates was expressed as follows: no biofilm formation (optical density test < optical density control), mild biofilm formation in the cases of (optical density control < optical density test < 2 optical density control), moderate biofilm formation in the cases of 2 optical denisty control < optical density test < 4optical density control), and strong biofilm formation in the cases of (4. optical density control < optical density test) [34,35].

### 2.6. Statistical Analysis

GraphPad Prism version 9 was used to analyze the data (La Jolla, CA, USA). The MAR index, biofilm formation, prevalence, and antibiotic resistance of *Aeromonas* isolates were all presented as percentages. The data were analyzed using simple descriptive statistics such as frequency distribution and percentages. The prevalence of infection was calculated by using the following formula: prevalence of infection (%) = no. of infected fish/total no. of examined fish

## 3. Results

A total of 42 isolates (12.2%) were characterized regarding cell morphology and Gram stain. All strains were rod-shaped, Gram-negative, oxidase-negative, and catalase-positive. The biochemical profile of the isolates, obtained by API20E, is shown in Figure 1. Fish sellers’ hands (12 isolates, 23.5%), tilapia (16 isolates, 28.8%), and mugil (14 isolates, 36.9%) were the sources of the contaminated samples; patient stool was not a source. (Table 2).

Based on the chemical characterization and molecular assays, 11 isolates with biochemical confirmation underwent amplification of the 16S rRNA gene specific to *Aeromonas* spp. The PCR primers used were 935 base pairs (bp) long. Figure 2A shows the confirmed identification of *Aeromonas* spp. (11 isolates, 3.2%). Following that, these isolates were tested using PCR for the 16S rRNA gene specific to *A. hydrophila* with an amplicon size of 625 base pairs (bp), which revealed that four isolates (2.1%) were positive (Figure 2B). Figure 2C illustrates that two isolates (50%) of the four identified *A. hydrophila* isolates were positive for the *aer* gene. These two isolates were from the tilapia and mugil fish samples. Conversely, the *hly* gene was found in 25% of the identified *A. hydrophila* isolates (Table 3).

The antibiotic susceptibility of the isolated and identified *Aeromonas* spp. was examined against different antibiotic groups. The sensitivity of the *Aeromonas* isolates to the tested antibiotic groups is shown in Table 4 as follows: cephalosporin, ceftriaxone (n = 2, 18.1%), cefotaxime (n = 2, 18.1%), ceftazidime (n = 2, 18.1%), cefixime (n = 2, 18.1%), carbapenem, imipenem 10 (n = 11, 100%), aminoglycoside, gentamycin (n = 9, 81.8%), tobramycin (n = 2, 18.1%), kanamycin (n = 3, 27.2%), streptomycin (n = 9, 81.8%), t, tetracycline (n = 0, 0%), quinolones, ciprofloxacin(n = 0, 0%), norfloxacin (n = 4, 36.3%), nalidixic acid (n = 3, 27.2%), phenicols, and chloramphenicol (n = 10, 90.9%) (Table 4). It is shown that the highest MAR index was recorded for mugil viscera (64.2%) against CFM, TE, CIP, CRO, CTX, TOB, K, S, and NA antibiotic groups followed by 57.1% against antibiotic classes, namely, CFM, TE, CIP, NA, CRO, CAZ, TOB, and K, for the hand swabs of *Aeromonas* isolates. On the other hand, 14.2% of mugil viscera *Aeromonas* isolates recorded MAR index of 0.142 against CFM, TE. SO, the highest MAR index was recorded by Fish samples including tilapia and mugil viscera. Additionally, multidrug resistance profile was shown and an isolate was considered MDR if it is resistant to at least one agent in three or more antimicrobial categories (Table 5).

The biofilm formation capacity by the confirmed *Aeromonas* isolates was tested and it was indicated that 2 isolates (18.1%) out of 11 confirmed isolates were biofilm producers. One of them was (9.1%) strong biofilm producer which was isolated from tilapia muscle, while the other 9 isolates (81.8%) were non- biofilm producers. (Table 6).

## 4. Discussion

In developing coastal nations, aquaculture is regarded a good source of animal protein that is suitable for human consumption [36]. The Nile tilapia is one of the most widely cultivated freshwater fish in Egypt and is regarded a crucial species in commercial fisheries [37]. Naturally infected Nile tilapia are frequently vulnerable to one or more stressors, such as hard handling, crowding, starvation, and high free ammonia (NH3) levels [38]. Fish represent an important source of vitamins, proteins, minerals, and unsaturated fatty acids [39]. The current study is very important since fish production is a main industry in Egypt and greatly affects its economy [40]. The most common bacterial pathogens infecting fish are *Aeromonas* spp., particularly *A. hydrophila,* which has been correlated with human illnesses, especially gastroenteritis, skin infections, respiratory infections, and septicemia [41].

Depending on the morphological and biochemical identification of the bacterial isolates mentioned in the Materials and Methods section, nine *Aeromonas* spp. were recovered from fish samples, and two *Aeromonas* spp. were recovered from fish sellers’ hands. In addition, four *A. hydrophila* species were obtained from fish samples. These findings match those of Sirijan Santajit et al. [33]. *Aeromonas* spp. were found in tilapia, mugil fish, fish sellers’ hands, and patient stool samples collected from Zagazig City, Sharkia Governorate, Egypt. In this study, the prevalence of *Aeromonas* in tilapia, mugil, stool, and hand swabs was investigated, and indicated that the area of study is thought to be a potential source of infectious diseases and fish contamination. [37]. This finding is in agreement with that of Rahimi et al., who reported a high prevalence of *A. hydrophila* in fish samples in Iran with a percentage of 19.5% [41]. In addition, Castro-Escarpulli et al. [42] recorded a prevalence of 32.8% of *Aeromonas* spp. isolates obtained from frozen fish. Morshedy et al. [1] found a greater isolation rate of 36% in cultured Tilapia nilotica. In the current study, the prevalence of *A. hydrophila* was 1.2% in tilapia and mugil viscera. Our findings are consistent with those of El-Ghareeb et al. [43], who isolated *A. hydrophila* from tilapia viscera, and with those of Ashiru et al. [44] in Nigeria, who reported no isolates from tilapia viscera. Our results could be explained by the microorganism’s pervasiveness and opportunistic behavior in the aquatic environment, as well as the fact that it coexists normally with other flora in fish intestines [45].

The present study showed that no *Aeromonas* spp. were recovered from patient stool samples. Our results coincide with those of the study conducted by Borchardt et al. [46], who reported that only 0.7% of *Aeromonas* spp. were recovered from stool samples of gastroenteric patients with watery diarrhea in the United States. While our results disagree with those reported by Tahoun et al. [47], who noticed a higher isolation rate (18.8%) from stool samples, our results indicated that fish sellers’ hands might represent a source of *Aeromonas,* and these results disagree with those of Ahmed et al. [39], who reported that no isolates were recovered from hand swab samples.

The variations in the prevalence of *Aeromonas* species could be attributed to presence of different species, sample conditions (such as time and location), geographic location, post-capture contamination, water type, fish species, handling, and manipulations during capture, storage, and transportation. This concurs with Hafez et al. [48].

The widespread usage of antibiotics has become rather typical as a result of the fishing industry’s continual expansion. Farmers have always employed various antibiotics to stop and cure pathogenic bacterial infections in fish in order to boost production [9]. Antibiotic-resistant strains have emerged all throughout the world as a result of the continuous and broad use of antibiotics by humans. Therefore, the antibiotic susceptibility of *Aeromonas* spp. to 16 antibiotics (six classes) was investigated. All *Aeromonas* spp. showed amoxicillin and cephalosporin resistance, which could be attributed to the lactamase enzyme produced by *Aeromonas* spp. via the expression of chromosomal lactamases [49]. Additionally, our results showed the higher resistance of *Aeromonas* spp. to cefixime, tetracycline, ciprofloxacin, and nalidixic acid, which is in agreement with the findings of Morshedy et al. [1] and Dhanapala et al. [50]. In agreement with our study, Ashuri et al. [44] reported a similar resistance to tetracyclin, while in contrast, Sarder et al. [51] reported a lower resistance (6.3) to ciprofloxacin. High resistance to ciprofloxacin and cefixime was reported by Sirijan Santajit et al. [32]. These findings suggest that antibiotic treatment might become useless in the near future, which is related to the extensive use of these antimicrobials for enhancing the growth of fish in aquatic environments, which in turn has resulted in an increase in the resistance patterns that could be transmitted to humans and animals [14,16,52]. The most effective antibiotics against all *Aeromonas* spp. were imipenem and chloramphenicol. This result concurs with that of Ahmed et al. [39], who reported 100% susceptibility to imipenem. Meanwhile, 50% resistance to imipenem has been reported by others [42]. Furthermore, several studies have reported *Aeromonas* spp. sensitivity to chloramphenicol [23].

The genus *Aeromonas* has been identified as an indicator bacterium for antimicrobial resistance in an aquatic ecosystem in previous research that focused on the antimicrobial resistance of other bacterial isolates [53,54]. Our results were consistent with those of earlier studies on the antibiotic resistance of *A. hydrophila* [52,53].

In this study, most *Aeromonas* spp. were resistant to more than three antibiotics, which revealed that the multiple antibiotic resistance (MAR) index was high, which confirmed the multidrug resistance and a high-risk contamination source. The high rates of multidrug resistance levels in the isolates collected in this study are alarming. Our result is in line with that of other research [54] that revealed a MAR index of 0.11 to 0.88 with an average of 0.489 or higher, revealing that fish represent a risky source of contamination [23].

Since virulence factors associated with extracellular products are crucial for the translocation in the epithelium, the presence of virulence-gene-positive *A. hydrophila* strains poses a serious risk to the public’s health [55]. According to certain research studies, the potential for predicting disease is correlated with the number of virulence genes present. The presence of virulence genes is mostly associated with the pathogenicity of *A. hydrophila*. Adhesins, hemagglutinins, and a number of hydrolytic enzymes are involved in the pathogenesis of *Aeromonas* spp. Hemolysin and aerolysin are implicated in these [56]. The aerolysin gene (*aer A*), which exhibits enterotoxic effects, is present in 50% of our strains in the current study. While other research [57,58] showed a greater frequency of aerolysin genes with a percentage of 70–100%, Yogananth et al. [59] obtained a result that was comparable. A hemolytic enterotoxin known as hemolysins (*hlyA*) was found to be present in 25% of the *A. hydrophila* isolates used in our analysis. This supports the conclusions made by Ahmed et al. [39], who reported that 28% of the examined *A. hydrophila* isolates in the Damietta Governorate, Egypt, were positive for *hlyA* gene. The *HLA*-*A* gene is expressed at higher levels, ranging from 30% to 100%, according to other studies [56,58]. *Aeromonas hydrophila* was found in fish farms in the East Delta with a higher percentage of *hlyA* genes (50%) than the percentages reported in our study [60]. *aerA* was found in *A. hydrophila* isolated from fish farms in the East Delta (100%) [61]. In addition, Hoel et al. [62] reported a higher prevalence of hemolysin-encoding genes (*hlyA* and *aerA),* with percentages of 99% and 98%, respectively. Meanwhile, Hafez et al. [48] reported similar results to our findings, with 20% for the aerolysin gene (*aerA*). Furthermore, the presence of multiple virulence factors may contribute to *A. hydrophila* pathogenicity, posing a public health risk [63].

Biofilms are structures formed by bacteria that increase their virulence and pathogenic capacity. *Aeromonas* bacteria form biofilms as a response to specific environmental circumstances [64]. Such biofilm structures enable virulence genes and surface antigens to be hidden in these structures and therefore enable bacterial resistance to antimicrobial agents such as antibiotics [65]. *Aeromonas* spp.’s prevalence in aquatic environments may be related to the presence of biofilm structures, and their presence in marine foods is a major source of human and animal infection. In the current study, biofilm production was reported in only two isolates of the tested *Aeromonas* spp. Biofilm formation has been documented by previous studies [34,66], which have reported similar findings to our study. Additionally, the capacity of bacteria to form biofilms boosts their antibiotic resistance and enhances their propensity to produce chronic infections.

Fish and shellfish are regarded as high-risk sources of bacteria that are resistant to several drugs and may be consumed by humans and animals or spread through improper handling. In order to provide information regarding the amount of contamination in the research area, it is crucial to continuously monitor the prevalence, antibiotic sensitivity, and biofilm production of such strains. Control techniques could be performed in order to prevent financial losses and serious diseases, depending on the information presented.

## 5. Conclusions

This study is an initial inquiry that will be followed by a more thorough analysis to assess the risk posed by the presence of pathogenic *Aeromonas* spp. in fish samples, which constitutes the basis of our country’s food supply. Using the methods employed, a sizable population of *Aeromonas* spp. was discovered. The identified *Aeromonas* spp. have a significant resistance profile to the tested antibiotics, which makes it more challenging to treat these bacteria. The risk of their transfer to humans through the food chain is further confirmed by the PCR discovery of the *hylA* and *aerA* genes in the examined *A. hydrophila*. In accordance with the information given, control techniques might be put into place to prevent the high-risk contamination of a particular location and therefore safeguard people from multidrug-resistant strains that might be spread through the food chain or improper handling.

## Figures and Tables

**Figure 1 biology-12-00421-f001:**
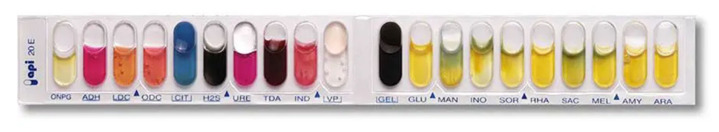
API-20 E biochemical profile of the *Aeromonas* isolates.

**Figure 2 biology-12-00421-f002:**
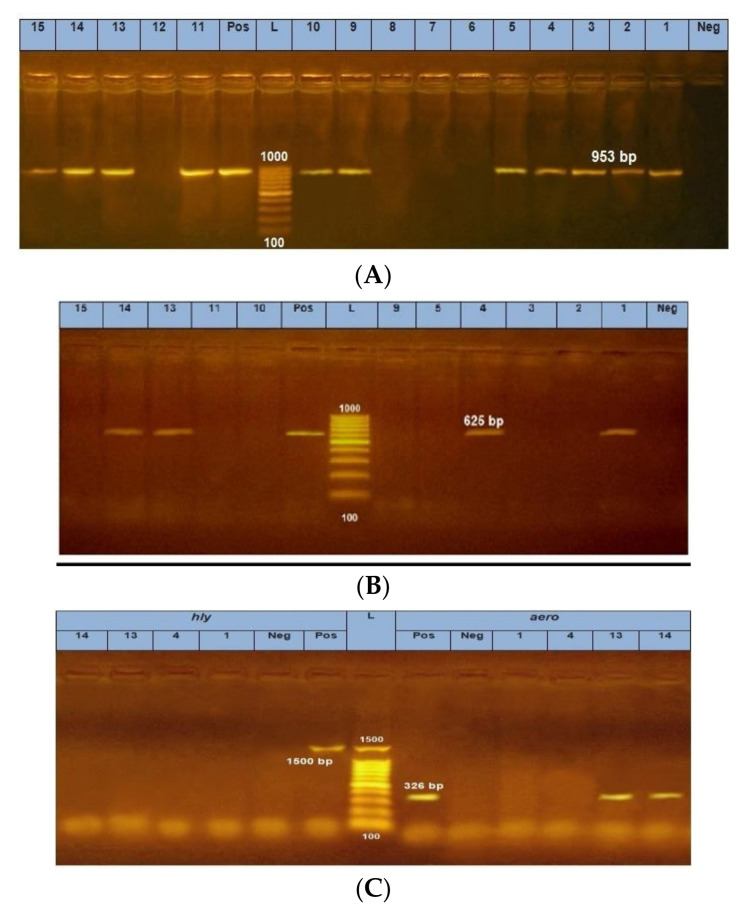
(**A**) Representation of polymerase chain reaction (PCR) positives for 16S rRNA of the genus *Aeromonas* with an amplicon size of 953 bp. L: 100 bp ladder; lanes 1–5, 9–11, and 13–15: positive samples; lanes 6–8 and 12: negative samples; lane Pos: positive control. (**B**) Representation of polymerase chain reaction (PCR) positives for 16S rRNA of genus *Aeromonas hydrophila* with an amplicon size of 625 bp. L: 100 bp ladder; lanes 1, 4, 13, and 14: positive samples; lanes 2, 3, 5, 9, 10, 11, and 15: negative samples; lane pos: positive control; lane neg: negative control. (**C**) Representation of polymerase chain reaction (PCR) positives for 16S rRNA of *aer* and *hly* gene amplification of *Aeromonas hydrophila* isolates. L: 100 bp ladder; for the *aer* gene lanes (13 and 14): positive samples. Lane (1 and 4): negative sample. Lane Pos: positive control and lane Neg: negative control. For the *hly* the, lanes 1, 4, 13, and 14: negative samples. Lane Pos: positive control and lane Neg: negative control.

**Table 1 biology-12-00421-t001:** Primers used for molecular identification by PCR.

Gene	Primer	Length (bp)	Reference
16S rRNA (genus-specific)		953	[20]
	F; 5′- CTA TGA AAA AAC TAA AAA TAA CTG -	′3	
	R; 5′- CAG TAT AAG TGG GGA AAT GGA AAG -	′3	
16S rRNA (species-specific)		625	[24,25]
	F; 5′- CAC AGC CAA TAT GTC GGT GAA G -′3		
	R; 5′- GTC ACC TTC TCG CTC AGG C--′3		

**Table 2 biology-12-00421-t002:** Incidence (%) of *Aeromonas* spp. isolated from fish and human samples.

Types of Samples		No. of Examined Samples	No. of Infected Samples (Positive Samples)	Percentage of Positivesamples
Tilapia: (n = 160)	Viscera	57	6	10.5%
	Liver	57	8	14.0%
	Muscles	46	2	4.3%
Mugil: (n = 105)	Viscera	38	8	21.0%
	Liver	25	1	4.0%
	Muscles	42	5	11.9%
Hand swabs (n = 51)		51	12	23.5%
Patient Stool (n = 27)		27	0	0
Total		343	42	12.2%

**Table 3 biology-12-00421-t003:** Distribution of molecularly identified *Aeromonas* spp., *A. hydrophila*, and virulence genes in the tested samples.

Code of Isolates	Source of Isolates	Molecularly Identified *Aeromonas* spp.	Molecularly Identified *A. hydrophila*	*Hly* Gene	*aer* Gene
Aer 1	Tilapia V *	+	+	-	-
Aer 2	Tilapia V	+	-	-	-
Aer 3	Mugil V *	+	-	-	-
Aer 4	Mugil V	+	+	-	-
Aer 5	Mugil M *	+	-	-	-
Aer 6	Mugil M	-	-	-	-
Aer 7	Mugil M	-	-	-	-
Aer 8	Mugil L *	-	-	-	-
Aer 9	Mugil V	+	-	-	-
Aer 10	Hand swabs	+	-	-	-
Aer 11	Hand swabs	+	-	-	-
Aer 12	Hand swabs	-	-	-	
Aer 13	Tilapia V	+	+	-	+
Aer 14	Mugil V	+	+	-	+
Aer 15	Tilapia M *	+	-	-	-

Aer: *Aeromonas* isolate Tilapia, V *: Tilapia viscera, Tilapia M *: Tilapia muscles, Mugil V *: Mugil viscera, Mugil L *: Mugil liver, Mugil M *: Mugil muscles, *hly*: hemolysin gene, *aer*: aerolysin gene.

**Table 4 biology-12-00421-t004:** Antibiotic suceptibilty of molecularly identified *Aeromonas* spp. against 16 antibiotics.

Antibiotics	*Aeromonas* spp. Isolates (no = 11) *RI * S **		
Ceftriaxone (CRO 30)	4 (36.4%)	5 (45.4%)	2 (18.2%)
Cefotaxime (CTX 30)	2 (18.2%)	7 (63.6%)	2 (18.2%)
Ceftazidime (CAZ 30)	3 (27.3%)	6 (54.5)	2 (18.2%)
Cefixime (CFM 5)	9 (81.8%)	-	2 (18.2%)
Imipenem (IPM 10)	-	-	11 (100%)
Gentamycin (CN 10)	-	2 (18.2%)	9 (81.8%)
Tobramycin (TOB 10)	5 (45.4%)	4 (36.4%)	2 (18.2%)
Kanamycin (K 30)	3 (27.3%)	5 (45.4%)	3 (27.3%)
Streptomycin (S 10)	1 (9.1%)	1 (9.1%)	9 (81.8%)
Tetracycline (TE 30)	10 (90.9%)	1 (9.1%)	-
Ciprofloxacin (CIP 5)	9 (81.8%)	2 (18.2%)	-
Norfloxacin (NOR)	2 (18.2%)	5 (45.4%)	4 (36.4%)
Nalidixic acid (NA 30)	5 (45.4%)	3 (27.3%)	3 (27.3%)
Chloramphenicol (C)	-	1 (9.1%)	10 (90.9%)

* R: resistant, I: intermediate, S: sensitive. CIP: ciprofloxacin; CFM: cefixime; S: streptomycin; TOB: tobramycin; IPM: imipenem; CN: gentamycin; NOR: norfloxacin; C: chloramphenicol; TE: tetracycline; CRO: ceftriaxone; CTX: cefotaxime; K: kanamycin; NA: nalidixic acid; CAZ: ceftazidime.

**Table 5 biology-12-00421-t005:** Distribution of antibiotic resistance (%) of *Aeromonas* spp. isolates and MAR index.

Bacterial Code	Isolates Source	No. of Resistant Antibiotics	Resistance Profile	Number of Antibiotic Classes	MAR Resi Index	Stance Level
Aer1	Tilapia Viscera	4	CFM, TE, CIP, NOR.	2	0.285	**DR**
Aer2	Tilapia Viscera	3	CFM, TE, CIP.	2	0.214	**DR**
Aer3	Mugil Viscera	2	CFM, TE.	2	0.142	**DR**
Aer4	Mugil Viscera	4	CFM, TE, CRO, CAZ	2	0.285	**DR**
Aer5	Mugil Muscles	3	TE, CIP, TOB	2	0.214	**DR**
Aer9	Mugil Viscera	5	CFM, TE, CIP, CAZ, TOB	3	0.357	**MDR**
Aer10	Hand swabs	5	TE, CIP, NOR, NA, TOB	2	0.357	**DR**
Aer11	Hand swabs	8	CFM, TE, CIP, NA, CRO, CAZ, TOB, K	3	0.571	**MDR**
Aer13	Tilapia Viscera	6	CFM, CIP, CRO, CTX, K, NA	3	0.428	**MDR**
Aer14	Mugil Viscera	9	CFM, TE, CIP, CRO, CTX, TOB, K, S, NA	3	0.642	**MDR**
Aer15	Tilapia Muscles	4	CFM, TE, CIP, NA	2	0.285	**DR**

CIP: Ciprofloxacin; CFM: Cefixime; S: Streptomycin; TOB: Tobramycin; IPM: Imipenem; CN: Gentamycin; NOR: Norfloxacin; C: Chloramphenicol; TE: Tetracycline; CRO: Ceftriaxone; CTX: Cefotaxime; K: Kanamycin; NA: Nalidixic acid; CAZ: Ceftazidime; DR, Drug resistant; MDR, Multidrug resistant.

**Table 6 biology-12-00421-t006:** Biofilm production by *Aeromonas* spp. using microtiter plate method.

Bacterial Code	Source of Isolates	Mean OD ± SD	Mean * ± SD	Degree of Biofilm
Control		0.103 ± 0.01 ^c^		
Aer 1	Tilapia Viscera	0.124. ± 0.02 ^c^	0.021 ± 0.021	Non
Aer 2	Tilapia Viscera	0.126 ± 0.007 ^c^	0.023 ± 0.007	Non
Aer 3	Mugil Viscera	0.108 ± 0.005 ^c^	0.005 ± 0.005	Non
Aer 4	Mugil Viscera	0.119 ± 0.012 ^c^	0.016 ± 0.012	Non
Aer 5	Mugil Muscle	0.117 ± 0.013 ^c^	0.014 ± 0.013	Non
Aer 9	Mugil Viscera	0.106 ± 0.002 ^c^	0.003 ± 0.002	Non
Aer 10	Hand Swabs	0.149 ± 0.004 ^c^	0.046 ± 0.004	Non
Aer 11	Hand Swabs	0.155 ± 0.009 ^c^	0.052 ± 0.009	Non
Aer 13	Tilapia Viscera	0.135 ± 0.029 ^c^	0.032 ± 0.029	Non
Aer 14	Mugil Viscera	0.27 ± 0.009 ^b^	0.167 ± 0.009	Weak
Aer 15	Tilapia muscles	0.701 ± 0.089 ^a^	0.598 ± 0.089	Strong

S: strong biofilm; M: moderate biofilm; W: weak biofilm; Non: non biofilm producer; SD: standard deviation. Mean *: the mean optical density (OD) value obtained from the media control well was deduced from all the test OD values, considered as an index for bacteria capable of forming biofilm. ^a,b,c^ Means with different superscripts are statistically different according to Duncan’s multiple range test.

## Data Availability

Not applicable.

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
