# Peer review of "Antibiotic Resistance, Virulence Gene Detection, and Biofilm Formation in Aeromonas spp. Isolated from Fish and Humans in Egypt"

_biology, 2023, doi:10.3390/biology12030421_

Round 1
Reviewer 1 Report (New Reviewer)
In this study, the authors screened fish and human samples to check the presence of Aeromonas spp. isolates and to determine their antimicrobial resistance, virulence, and biofilm formation profiles of those isolates. Although the significance of the current study is high for both fish and human health, this study has a few serious flaws which must be addressed and corrected to improve the quality of the study. Please find and check my comments below:
Overall comments
· Your abstract is very poor. Please improve it following these steps: background/introduction > Objectives > methodology > results > conclusion. Also, please don’t use all the names of antibiotics in your abstract. And please try to show the important outcomes, e.g., focus on the resistance profiles instead of sensitive profiles.
· This manuscript has a lot of punctuation and space errors. Please correct them throughout the manuscript.
· “spp.” must not be italic. Please correct it throughout the manuscript.
· According to EUCAST guidelines, Aeromonas species show intrinsic resistance to amoxicillin/ clavulanic acid and amoxicillin. Why did you use them in your study? There is no sense to use them for Aeromonas spp. You must omit them from this study. I believe the results of AMR should be drastically changed after omitting these two antibiotics from your study.
· How could you find the zone diameter for amoxicillin/ clavulanic acid in CLSI 2015.m45? I believe there is no data for that antibiotic in the CLSI guidelines. Please clarify it.
· Please use the italic form for all genes’ names.
· For the number one to ten, please don’t use them as numerical numbers. Please correct it where needed.
· Discussion: Please improve your discussion section. Please don’t show all the results here. Just summarize your key findings in concise language.
Other comments
Title: Please change the title to “Antibiotic resistance, virulence gene detection, and biofilm formation in Aeromonas spp. isolated from fish and humans in Egypt”. You don’t need to use the word “prevalence”, “capacity” or “samples” in the title.
Introduction: Please provide the knowledge gap and/or justification of your study before the objectives of the present study.
Keywords: All the keywords are available in the title. Please include a few relevant keywords which are not present in the title.
Line 84: Objectives should be in past form.
Line 108: Please write “Twenty-five” instead of “25” here.
Line 140-144: Please arrange the sequence of these two primers in Table 1.
Line 155-163: Please provide the concentration for all the antibiotic disks here. You have shown them in the results section. Please move them here. Also, the data for ciprofloxacin is available in CLSI 2015.m45, why did you use EUCAST guidelines for that antibiotic? In addition, I couldn’t find any zone diameter data for streptomycin in CLSI 2017. Please clarify them and provide proper references.
Line 166: Please use the appropriate reference for calculating the MAR index. You should use the following reference: https://doi.org/10.1128/aem.46.1.165-170.1983
Line 193-195: You have already mentioned the definition of the MAR index in section 2.4. Please omit this from here.
Line 197-199: Please omit this sentence. You have already mentioned the number of samples in the materials and methods section.
Line 212-213: Please omit this sentence. This sentence is for materials and methods. You shouldn’t use it in the results section.
Line 242-248: Please don’t use the short of antibiotics’ name in the manuscript. This format is suitable only for tables or figures. However, the use of a short form is okay for showing the resistance pattern.
Line 277-278: Please omit this sentence. This sentence is for materials and methods. You shouldn’t use it in the results section.
The first paragraph of the discussion section: Please minimize the size of this paragraph. It seems like you are writing the introduction section again.
Author Response
"Please see the attachment."

Reviewer 2 Report (New Reviewer)
The manuscript includes the characteristics of Aeromonas spp. regarding the antibiotic resistance, presence of virulence genes and biofilm formation. The authors examined both fish and human samples, using appropriate research methods. Despite the fact that the authors presented in detail data, the manuscript is unclear in some places and requires revision.
Comments:
Line 23: The authors should indicate unequivocally that the samples included fish tissues and organs, and human samples of both skin swabs and faecal samples.
Does n mean the number of people/animals or the number of samples?
Line 90-91: The authors described the liver samples in the results. It should be add at the material and methods.
Line 93: The authors should write what the condition of the fish was.
Line 108: Which samples weighed 25g? how were the swabs inoculated? Please clarified.
Line 100: The authors should clarified that samples were collected from human.
Line 116-117 (The colonies were….) should be move to line 112.
Line 122: The authors should clarified which species specific primers were used.
Results: The authors used API 20E kit to identification of bacteria. Please add the biochemical characteristics of bacterial strains or their biochemical profile.
Table 2: Please clarified what means: No. of infected samples. Did the authors mean no. of positive samples?
Discussion: Requires revision. There are repeated results in many places and discussion sometimes is not clear.
Author Response
"Please see the attachment."

Round 2
Reviewer 1 Report (New Reviewer)
The authors tried to address my comments but couldn't do them properly. They need to address the following comments:
> I believe this manuscript needs English correction. You used several complicated sentences which won't be clear to readers.
> The scientific name of the organism should be in italics. Please correct it throughout the manuscript.
> Please minimize the background of the simple summary of your manuscript.
Line 51: bracket missing
Line 56: use "and" after "25%"
Line 57: It should be 14 antibiotics; please recheck it.
Line 58: You've mentioned that you omitted amoxicillin/clavulanic acid and amoxicillin, but you still showed the results of amoxicillin/clavulanic acid here. Also, I've found one of these antibiotics throughout the manuscript. Please correct them properly.
Also, please don't show any published articles for guidelines to perform antibiotic susceptibility tests (using disk diffusion test). You should always follow the approved guidelines such as CLSI, EUSCAST, CDC, etc.
Line 68: You don't need to mention "MAR" here.
Line 112: It should be "food"
Knowledge gap means like did you perform your study for the first time in your country or did you design your study to provide up-to-date information about this organism in raw fish in your country. Please provide this information before the objectives of your study.
Line 114: "was" should be "were"
Line 121-123: Please provide this information "57 samples 121 from viscera, 57 liver samples and 46 muscle samples" and " 38 viscera samples, 25 liver samples and 42 muscle samples" in the bracket.
Line 125-131: Please make this information simple. It seems complicated.
Line 137-140: Please paraphrase this sentence. It seems complicated.
Table 3: "Hly" should be "hly" and "Aer" should be "aer"
Line 301: The class of chloramphenicol should be only "phenicols"
Line 304-306: Please omit this sentence
Line 307: Nine isoates!!! What do you mean by that? Please check these kinds of errors and correct them.
Moreoer, Lin 293-318: Show these results in a concise form. All the results are available in the table.
Table 4: Please check the percentages of R, I, and S by adding them to make 100%. Also, you don't need to mention the abbreviation of antibiotics and their concentration here because you have already mentioned them under the table and in the materials and methods sections, respectively.
Line 323: Amoxicillin???
Table 5: Is it important to show the resistance rate here? You have already shown the MAR indices. In addition, the results of multidrug-resistant (MDR) isolates are more important than MAR indices. Why didn't you show the MDR profiles of the isolates? Please show it.
Also, please check the MAR indices, as you needed to omit two antibiotics.
Discussion: Please check the discussion section properly for space, punctuation, and the scientific name of the organism.
Author Response
- I believe this manuscript needs English correction. You used several complicated sentences which won't be clear to readers.
- We appreciate the reviewer for his/ her suggested corrections.
- The English language of the manuscript is reviewed and the grammar content, typos errors and complicated sentences are simplified by a native English speaker.
- The scientific name of the organism should be in italics. Please correct it throughout the manuscript.
- It was corrected throughout the manuscript
- Please minimize the background of the simple summary of your manuscript.
- It was minimized.
- Line 51: bracket missing.
- It was added.
- Line 56: use "and" after "25%".
- It was added
- Line 57: It should be 14 antibiotics; please recheck it.
- It was done.
- Line 58: You've mentioned that you omitted amoxicillin/clavulanic acid and amoxicillin, but you still showed the results of amoxicillin/clavulanic acid here. Also, I've found one of these antibiotics throughout the manuscript. Please correct them properly.
- It was corrected throughout the manuscript.
- Also, please don't show any published articles for guidelines to perform antibiotic susceptibility tests (using disk diffusion test). You should always follow the approved guidelines such as CLSI, EUSCAST, CDC, etc.
- It was corrected and a reference for CLSI was added in the reference list
- Line 68: You don't need to mention "MAR" here.
- It was deleted.
- Line 112: It should be "food"
- It was done.
- Knowledge gap means like did you perform your study for the first time in your country or did you design your study to provide up-to-date information about this organism in raw fish in your country. Please provide this information before the objectives of your study.
- It was added.
- Line 114: "was" should be "were"
- It was done.
- Line 121-123: Please provide this information "57 samples 121 from viscera, 57 liver samples and 46 muscle samples" and " 38 viscera samples, 25 liver samples and 42 muscle samples" in the bracket.
- The brackets were added.
- Line 125-131:Please make this information simple. It seems complicated.
- It was simplified.
- Line 137-140:Please paraphrase this sentence. It seems complicated.
- The sentence was paraphrased.
- Table 3:"Hly" should be "hly" and "Aer" should be "aer"
- It was done.
- Line 301:The class of chloramphenicol should be only "phenicols"
- It was done.
- Line 304-306:Please omit this sentence
- It was omitted.
- Line 307: Nine isoates!!! What do you mean by that? Please check these kinds of errors and correct them.
- I mean the number of isolates which showed resistance to , however it was corrected according to the reviewer comments
- Moreoer, Lin 293-318:Show these results in a concise form. All the results are available in the table.
- The results were concised according to the reviewer comment.
- Table 4: Please check the percentages of R, I, and S by adding them to make 100%. Also, you don't need to mention the abbreviation of antibiotics and their concentration here because you have already mentioned them under the table and in the materials and methods sections, respectively.
- The percentages were checked and corrected to make 100%
- The abbreviation and concentrations were required by the reviewer No.2 to be added in the table so, I added them according to the comments of reviewer No.2
- Line 323: Amoxicillin???
- It was deleted.
- Table 5: Is it important to show the resistance rate here? You have already shown the MAR indices. In addition, the results of multidrug-resistant (MDR) isolates are more important than MAR indices. Why didn't you show the MDR profiles of the isolates? Please show it.
- Thanks dear reviewer, the resistant rate column was deleted from the table .
- The aim of the current study was to determine MAR index only so, We donot add MDR.
- Thanks a lot my dear reviewer and multidrug resistance profile, and number of antibiotic classes were added in the table as an isolate was considered MDR if it is resistant to at least one agent in three or more antimicrobial categories.
- Also, please check the MAR indices, as you needed to omit two antibiotics.
- MAR indices were well checked after the two antibiotics omitted.
- Discussion: Please check the discussion section properly for space, punctuation, and the scientific name of the organism.
- Thanks a lot for this point and the discussion section was properly checked
Thank you

Reviewer 2 Report (New Reviewer)
I am satisfied with the correction of the manuscript and will recommend it be accepted for publication.
Author Response
- I am satisfied with the correction of the manuscript and will recommend it be accepted for publication.
- Thanks a lot dear reviewer and I really want to express my greetings and lot thanks to yours and I really grateful for your recommendation by the acceptance of our manuscript
Thank you

Round 3
Reviewer 1 Report (New Reviewer)
The authors addressed almost all of my comments. However, I have two comments (The 3rd one must be addressed properly) as follows:
1> Line 201, 297: "aer" and "hly" should be in italics.
2> Line 242-245: Please provide the full form of "ODs" and "ODnc" here.
3> Table 5: You didn't correct the MAR index. You have kept the previous one. You have to divide by 14 now. Please correct it and change the MAR indices results throughout the manuscript. Also, please provide the full form of antibiotic names under Table 5.
Author Response
- Response to reviewer No. 1.
- The authors addressed almost all of my comments. However, I have two comments (The 3rd one must be addressed properly) as follows:
- Thanks a lot dear reviewer and we appreciate the reviewer for his/ her suggested corrections and all of them were done.
- Line 201, 297: "aer"and "hly" should be in italics.
- It was done.
- Line 242-245: Please provide the full form of "ODs" and "ODnc" here.
- The full forms were provided.
- Table 5: You didn't correct the MAR index. You have kept the previous one. You have to divide by 14 now. Please correct it and change the MAR indices results throughout the manuscript. Also, please provide the full form of antibiotic names under Table 5.
- Thanks a lot for this point, the MAR indices results were calculated and corrected again in the table and also, throughout the manuscript.
- The full form of antibiotic names were also provided under Table 5.
Thank you

This manuscript is a resubmission of an earlier submission. The following is a list of the peer review reports and author responses from that submission.
Round 1
Reviewer 1 Report
Manuscript entitled “Incidence, antibiotic resistance, virulence genes detection and biofilm formation capacity by Aeromonas spp. isolated from fish and human samples in Egypt”. In this manuscript, the distribution of Aeromonas sp was investigated in fish and fish seller's hand samples. However, there are some problems in experimental design, the content of manuscript and writing. There are also many mistakes in the manuscript and the experimental results don't show good and new results. So I recommend that this manuscript cannot be accepted. The reviewer gives the following comments and suggestions.
1. The title word “incidence” is not appropriate. Aeromonas sp are widely distributed in aquatic environments and production, and Aeromonas sp are opportunistic pathogens of animals including humans.
2. The goals of manuscript are unclear.
3. The experiment design is too simple:
(1) The detection of stool samples seems not directly related to the pathogens of fish.
(2) Only 11 isolated strains were identified as Aeromonas spp in the manuscript, it is difficult to statistics and data analysis.
(3) The manuscript should explain the times, places, and quantity of collecting samples.
(4) The method of hand swab sample collecting should be described more clearly. (5) The 16s rRNA gene is inaccurate to identify bacterial species, please used more housekeeping genes (such as gyrB and rpoB) to identify bacterial species.
(6) More virulence genes should be detected in the manuscript, such as enterotoxin, ectoenzyme, transferase, and virulence genes should be detected in all isolated Aeromonas strains.
4. There are too many mistakes in the manuscript, please check all the manuscript. Page 2, paragraph 1 for example:
(1) Misuse italicize in line 1 to 7.
(2) “A. salmonicidae.” should delete “.” in line 7.
(3) The A. salmonicidae is non-motile bacterium, it cannot produce MAS.
(4) There is a lack of “[11]”.
(5) There needs a space after “fish and humans.” in line 15.
Author Response
- Response to reviewer No. 1.
- The title word “incidence” is not appropriate. Aeromonas sp are widely distributed in aquatic environments and production, and Aeromonas spare opportunistic pathogens of animals including humans.
- Thanks very much for this point, the title word incidence is changed to the title word '' Prevalence'' which is more accurate and represent the wide distribution of Aeromonas sp. in aquatic environments and the high population of people affected by those opportunistic pathogens.
- So the title of the manuscript changed to '' Prevalence, antibiotic resistance, virulence genes detection and biofilm formation capacity by Aeromonas spp. isolated from fish and human samples in Egypt.
- The goals of manuscript are unclear.
- The goal of this study is indicated at the end of the abstract ''This study aims to establish a baseline for monitoring and controlling the multidrug-resistant Aeromonas spp. and especially Aeromonas hydrophila in marine foods to protect humans and animals''. In addition to that, the aim of the study rewritten again at the end of the Introduction and became more clear and obvious.
- The experiment design is too simple:
- The detection of stool samples seemsnot directly related to the pathogens of fish.
- For this point, Aeromonas sp. are emerging human enteric pathogens and play an important role in causing gastroenteritis among humans and children especially in developing countries which is supported by many previous studies [Fernández-Bravo, A.; Figueras, M.J. An Update on the Genus Aeromonas: Taxonomy, Epidemiology, and Pathogenicity. Microorganisms 2020, 8, 129. [CrossRef] [PubMed]. In addition, Aeromonas spp. can be isolated from stool samples of healthy individuals and Several studies reported an association with acute and prolonged diarrhea with percentage ranging from 2% to 20% [Castaño-Rodríguez, N.; Underwood, A.P.; Merif, J.; Riordan, S.M.; Rawlinson, W.D.; Mitchell, H.M.; Kaakoush, N.O. Gut Microbiome Analysis Identifies Potential Etiological Factors in Acute Gastroenteritis. Infect. Immun. 2018, 86, e00060-18] & [Tarr, G.A.M.; Chui, L.; Lee, B.E.; Pang, X.-L.; Ali, S.; Nettel-Aguirre, A.; Vanderkooi, O.G.; Berenger, B.M.; Dickinson, J.; Tarr, P.I.; et al. Performance of Stool-testing Recommendations for Acute Gastroenteritis When Used to Identify Children With 9 Potential Bacterial Enteropathogens. Clin. Infect. Dis. 2018, 69, 1173–1182. [CrossRef] [PubMed]
In relation to our study , a more recent study supported our finding and found that up to 3.6% from the studied 50 stool samples were positive for Aeromonas sp. especially A. hydrophila, A. caviae, A. veronii, and A. eucrenophila and 42% of the samples were positive for Aeromonas sp. alongside other significant pathogens and the reference is [Grave, I.; Rudzate, A.; Nagle, A.; Miklasevics, E.; Gardovska, D. Prevalence of Aeromonas spp. Infection in Pediatric Patients Hospitalized with Gastroenteritis in Latvia between 2020 and 2021. Children 2022, 9, 1684. https://doi.org/10.3390/ children9111684
- Only 11 isolated strains were identified as Aeromonas spp in the manuscript, it is difficult to statistics and data analysis.
- For this point, we isolated 343 samples from fish and patients and only 11 isolates were proved to be positive for Aeromonas spp. and statics made in our study were only percentage of positive samples in relation to the total samples isolated which not needed complex statics or analysis. In addition, MAR index also not needed special calculations or statics as it represented the no. of resistant strains to the total no. of antibiotics. So, the calculations were all made as percentage and not needed complex statics. All of these results were clearly indicated in our manuscript.
- The manuscript should explain the times, places, and quantity of collecting samples.
- The answer on this point is given in the text in section [Materials and metods] in the 2.1. sample collection paragrah. There were 265 fish samples (quantity) gathered , including 160 tilapia (Oreochromis niloticus) and 105 mugil (Mugil cephalus), were collected during the different seasons of 2022 (time) from local fish markets in Sharkia Governorate (80 km north of Cairo), Egypt (place) and 51 hand swabs and 27 stool samples (quantity) were obtained in sterile alkaline peptone water after being collected under aseptic circumstances (APW, Oxoid CM1028). From the university hospitals in Zagazig City, Egypt (place).
- The method of hand swab sample collecting should be described more clearly.
Thanks a lot . the method was rewritten again and became more obvious.
- The 16s rRNA gene is inaccurate to identify bacterial species, please used more housekeeping genes (such as gyrBand rpoB) to identify bacterial species.
For this point, although gyrB and rpoB may be effective for the classification of Aeromonas spp., but 16s rRNA represents a powerful tool for the rapid and accurate identification of random collected species and many previous and recent studies depend on 16srRNA for the rapid identification of the isolated Aeromonas spp. and for the detection of hly A gene [Santajit S, Kong-ngoen T, Tunyong W, Pumirat P, Ampawong S, Sookrung N, Indrawattana N (2022) Occurrence, antimicrobial resistance, virulence, and biofilm formation capacity of Vibrio spp. and Aeromonas spp. isolated from raw seafood marketed in Bangkok, Thailand, Veterinary World, 15(7): 1887–1895] .In addition, These rapid DNA-based detection methods are simple, easy to perform and faster in identifying Aeromonas spp., thus are absolutely efficient for regular monitoring of Aeromonas spp. in a potential outbreak situation and this what we aim in our current study which is supported by a previous study of [Diyana-Nadhirah, K.P., and M.Y. Ina-Salwany. 2016. Molecular Characterization of 16S rRNA and Internal Transcribed Spacer (ITS) Regions of Aeromonas spp. Isolated from Cultured Freshwater Fishes in Malaysia.Int.J.Curr.Microbiol.App.Sci. 5(9): 431-440.]. Our results are in accordance with that documented in article by Aboyadak et al. (2015) who observed Aeromonas having 953 bp isolated from tilapia in Egypt which is another support for our results.
- More virulence genes should be detected in the manuscript, such as enterotoxin, ectoenzyme, transferase,and virulence genes should be detected in all isolated Aeromonas
For this point , although aeromonas spp. are considered as food borne pathogens but A. hydrophila is the most prevalent pathogen associated with gastroenteritis, septicemia and traveler’s diarrhea in humans and hemorrhagic septicemia in fish as supported by the previous study of [Lopatek, M., Wieczorek, K. and Osek, J. (2018) Antimicrobial resistance, virulence factors, and genetic profiles of Vibrio parahaemolyticus from seafood. Appl. Environ. Microbiol., 84(16): e00537–18.] so, we focused on it in our manuscript for its most prevalent pathogenicity.
Since the prevalence of the aer gene and hyl gene are the most common virulence genes of A. hydrophila which are mainly related to its hemolytic activity which is responsible for thtroenteritis and hemmorhagic symptoms in fish so, we focused on the molecular detection of these virulence genes and Similar studies have been conducted in other countries such as Canada, China, India, Malaysia and Spain (Wang et al., 2003; Pridgeon and Klesius, 2011; Zhou et al., 2013) which support our results. Some recent published data such as in case of Hossain et al. (2020), reported 80.8% hyl gene and Mansour et al. (2019) found 5/21 hyl genes from diseased fish of Kafr El-Shaikh Egypt. All of these recent studies supported our findings and why we focused on these virulence genes in our study.
- In addition, the goal of our study was not to study and detect all the virulence genes in all tested samples but our goal is to detect the virulence genes which are mainly related to the pathogenicity of Aeromonas spp. isolated from our local markets especially Aeromonas hydrophila which was proved to be the most prevalent pathogen in our study ; further work will be necessary to check such point.
- There are too many mistakes in the manuscript, please check all the manuscript. Page 2, paragraph 1 for example:
- We appreciate the peer and rigorous revision of the respected reviewer.
- The manuscript is reviewed with completely care and all the errors were corrected.
- Misuse italicize in line 1 to 7
- It is ok and is done
- “ salmonicidae.” should delete “.” in line 7.
- It is done, thanks a lot.
- The salmonicidae is non-motile bacterium, it cannot produce MAS.
- salmonicidae was deleted
- There is a lack of “[11]”.
- Thanks a lot, it was added.
- There needs a space after “fish and humans.” in line 15.
- It was done
Thank you
Reviewer 2 Report
This research is kindly study about the incidence and investigation for virulence genotype and biofilm formation of Aeromonas spp. isolated from fish and human specimens in Egypt.
- On page 2, please correct the italics of the words in the sentence, "Aeromonas spp. are significant contributors to the normal microbial flora in fresh- water and saltwater environments. They also pose a threat to humans because they can cause dangerous diseases like septic arthritis, gastroenteritis with diarrhoea, skin and soft tissue infections, meningitis, and bacteremia [8,9]. A. hydrophila transmits the zoonotic disease aeromoniasis to humans through the intake of tainted fish and water [10]."
- Please delete the full-stop symbol (.) between "salmonicidae" and "alongside" and correct for the space between sentences throughout the manuscript.
- Please add the country instead of the city of alkaline peptone water in Materials and Methods 2.2.
- In Materials and Methods 2.3, change "hly A" to "hylA".
- In Materials and Methods 2.4: "2x 108 CFU/mL" to "2x 108 CFU/mL"
- Please add the ethical approval for samples collected from the patients.
- Please provide more than just the percentage in statistical analysis part.
Author Response
- Response to reviewer No.2.
- This research is kindly study about the incidence and investigation for virulence genotype and biofilm formation of Aeromonas spp. isolated from fish and human specimens in Egypt.
- Thanks a lot for your kind comment and we appreciate the reviewer for his suggested corrections and all of them were done.
- On page 2, please correct the italics of the words in the sentence, "Aeromonas spp. are significant contributors to the normal microbial flora in fresh- water and saltwater environments. They also pose a threat to humans because they can cause dangerous diseases like septic arthritis, gastroenteritis with diarrhoea, skin and soft tissue infections, meningitis, and bacteremia [8,9]. A. hydrophila transmits the zoonotic disease aeromoniasis to humans through the intake of tainted fish and water [10]."
- Thanks a lot, all the italics were corrected.
- Please delete the full-stop symbol (.) between "salmonicidae" and "alongside" and correct for the space between sentences throughout the manuscript.
- It was deleted and all the spaces throughout the manuscript
- Please add the country instead of the city of alkaline peptone water in Materials and Methods 2.2.
- It was done and thanks a lot.
- In Materials and Methods 2.3, change "hly A" to "hylA".
- It was changed.
- In Materials and Methods 2.4: "2x 108 CFU/mL" to "2x 108 CFU/mL"
- It was done.
- Please add the ethical approval for samples collected from the patients.
- It was already sent to the biology editorial office by the date of 8/11/2022 and added in the manuscript Informed Consent Statement: "Written informed consent has been obtained from the patient(s) to publish this paper
- Please provide more than just the percentage in statistical analysis part.
- Thanks a lot , but for this point MAR index maily calculated as percentage and for biofilm formation, standard deviation was calculated and for antibiotic sensitivity , no additional statics was needed , we only used the percentage to indicate the resistance or sensitivity of isolates to the tested antibiotics.
Thank you
Reviewer 3 Report
Please find the attached comments. Also, Adding line numbers in the manuscript would have been very helpful in writing comments for the reviewers.
Incidence, Antibiotic resistance, virulence genes detection and biofilm formation capacity by Aeromonas spp. Isolated from fish and human samples in Egypt
Mahdy et al.,
Comments
It is a well written manuscript and has significance in terms of public health. However, please address the following comments. Adding line numbers in the manuscript would have been very helpful in writing comments for the reviewers.
Page 2: Change the font from italicized to normal in the first few lines of first paragraph
Page 2 : Correct the spelling of “transimetted”
Page 2 (2.1): It has been mentioned that, out of 265 fish samples, 51 were hand swabs, 27 were stool samples, how about the remaining samples? How many were viscera? How many were muscle samples? Etc.
Page 3 (2.1): Were all the samples collected in alkaline peptone water? Or only hand swabs and stool samples? Mention clearly
Page 3 (2.2): Was alkaline peptone water used for both sample collection and sample enrichment?
Page 4 (2.4) : Was the turbid broth streaked on MH agar? Or plated on MH agar plate for confluent lawn of bacterial growth?
Page 9 (4): In 2nd paragraph- Is it total of 11 Aeromonas isolates based on biochemical and morphological identification as mentioned in page 5. Only 9 isolates have been mentioned here
Page 11 (5): Conclusions derived from results of the study should be mentioned here. Please do not mention the methods used in the study to arrive at the conclusions (you have mentioned virulence genes were identified, biofilm formation was studied etc.)
Author Response
- Response to reviewer No.3
- It is a well written manuscript and has significance in terms of public health. However, please address the following comments.
- Thanks a lot for dear colleague. We appreciate the reviewer for his/ her suggested corrections and all the comments were done.
- Page 2: Change the font from italicized to normal in the first few lines of first paragraph
- It was changed
- Page 2 : Correct the spelling of “transimetted”
- It was corrected.
- Page 2 (2.1): It has been mentioned that, out of 265 fish samples, 51 were hand swabs, 27 were stool samples, how about the remaining samples? How many were viscera? How many were muscle samples? Etc.
- Thanks a lot for this point, but it was indicated that the total collected samples were 343 and out of 343, 265 samples were fish samples (160 tilapia, 105 mugil).
- For 160 Nile tilapia, it was indicated in table 2 on page 5, that 57 out of 160 were viscera, 57 were from liver and 46 were muscles. And from 105 mugil, 38 were viscera, 25 were liver and 42 were muscles.
- Also, I already added a line that indicated the total samples were 343 and out of 343, 265 were fish samples at the beginning of 2.1. paragraph in the ''Materials and methods section''.
- Page 3 (2.1): Were all the samples collected in alkaline peptone water? Or only hand swabs and stool samples? Mention clearly
- No, Fish samples were collected in sterilised polyethylene bags on ice but hand swabs and stool samples were obtained in sterile alkaline peptone water
- Page 3 (2.2): Was alkaline peptone water used for both sample collection and sample enrichment?
- Yes, APW was used for collection of hand swaps and stool samples and used for enrichment of all collected samples either fish samples or swaps and stool samples.
- Page 4 (2.4): Was the turbid broth streaked on MH agar? Or plated on MH agar plate for confluent lawn of bacterial growth?
- The turbid broth was streaked on MH agar.
- Page 9 (4): In 2nd paragraph- Is it total of 11 Aeromonas isolates based on biochemical and morphological identification as mentioned in page 5. Only 9 isolates have been mentioned here
- Thanks a lot, yes, there is a total of 11 Aeromonas isolates, 9 of them were from fish samples and two from fish sellers and it was corrected in the Discussion section.
- Page 11 (5): Conclusions derived from results of the study should be mentioned here. Please do not mention the methods used in the study to arrive at the conclusions (you have mentioned virulence genes were identified, biofilm formation was studied etc.)
- The conclusion rewritten again and became more clear and obvious.
Thank you